# Impact of Iran’s Forest Nationalization Law on Forest Cover Changes over Six Decades: A Case Study of a Zagros Sparse Coppice Oak Forest

**DOI:** 10.3390/s23020871

**Published:** 2023-01-12

**Authors:** Hadi Beygi Heidarlou, Abbas Banj Shafiei, Vahid Nasiri, Mihai Daniel Niţă, Stelian Alexandru Borz, David Lopez-Carr

**Affiliations:** 1Department of Forest Engineering, Forest Management Planning and Terrestrial Measurements, Faculty of Silviculture and Forest Engineering, Transilvania University of Brasov, Sirul Beethoven 1, 500123 Brașov, Romania; 2Forestry Department, Faculty of Natural Resources, Urmia University, Urmia P.O. Box 165, Iran; 3Faculty of Civil Engineering, Transilvania University of Brasov, Turnului 5, 900152 Brașov, Romania; 4Department of Geography, University of California Santa Barbara, Santa Barbara, CA 93106, USA

**Keywords:** change detection, conservation policies, declassified satellite photography, Google Earth Engine, historical aerial photos, machine learning

## Abstract

Forest nationalization policies in developing countries have often led to a reduction in local forest ownership rights and short- or long-term exploitative behaviors of stakeholders. The purpose of this research is to quantify the effect of Iran’s Forest Nationalization Law (FNL) in a part of Zagros Forest over a 68-year time period (1955–2022) using 1955 historical aerial photos, 1968 Corona spy satellite photography, and classification of multi-temporal Landsat satellite images. A past classification change detection technique was used to identify the extent and the pattern of land use changes in time. For this purpose, six periods were defined, to cover the time before and after the implementation of FNL. A 0.27% deforestation trend was identified over the period after the FNL. Dense and open forested area has decreased from 7175.62 ha and 68,927.46 ha in 1955 to 5664.26 ha and 59,223.38 ha in 2022. The FNL brought decisive changes in the legal and forest management systems at the state level, mainly by giving their ownership to the state. Accordingly, the FNL and the related conservation plans have not fully succeeded in protecting, rehabilitating, recovering, and developing the sparse Zagros Forest ecosystems, as their most important goals.

## 1. Introduction

Forests are among the most important components of sustainable development. Despite several ecological, economic, educational, and social advantages, they have been damaged consistently over time by various human activities, including policy and management changes, one of which is the destruction of vegetation as a result of land-cover and land-use change (LCLUC) [1]. LCLUC is the result of a complex interaction between physical, biological, and social factors, including policies and regulations adopted at the national, provincial, and regional levels [2,3]. LCLUC is mainly affected by the global economy, climate, demographic change, and local policies [4,5].

In Asia and developing countries, forest nationalization policies have often led to a reduction in local societies’ ownership of forests, their exploitative behaviors, which finally, caused deforestation [6]. Several studies have shown that legal and illegal activities (i.e., illegal logging and related trades) are crucial elements of the management and sustainable use of natural resources by local societies [7,8]. The collection of public property rights over goods includes the rights to use, administer, and conserve them [9]. Although some studies have suggested that legal rights are necessary for the long-term viability of locally-controlled forests [6], in some instances understanding the indigenous knowledge can help maintain those systems’ stability [10]. Long-term management incentives may also change when an individual or community’s ownership of a resource changes as a policy or law is implemented. For instance, if ownership of the entire forest is taken away from the public, exclusivity and security will likely result in little incentive to manage the forest [11]. In contrast, in the absence of transparency of property rights, the resources may be vulnerable to destruction due to free access [6]. Although transparent and equitable property rights of traditional indigenous forest dwellers are essential for a national forestry plan [12], many countries have adopted policies that reflect the ultimate property rights of forests such as Indonesia (Basic Forestry Law (1967)), India (1878 Indian Forest Act), Nepal (Private Forests Nationalization Act (1957)), Thailand (the Forest Preservation Act (1913)), Zambia (The Forests Act, 1973), Papua New Guinea (Forestry (Amendment) Act 1993), the United States (Forest Reserve Act of 1891), and Switzerland (Forest Act, ForA). When these laws were put into action, they immediately gave the government ownership over local users’ property. In essence, the local community is denied any legal title to the forests. Frequently, these modifications in ownership rights led to new incentive systems which were in the form of shifting accessibility standards and conditions [13].

Since the beginning of Iran’s public management of natural resources, various policies have been implemented. Some laws have been enacted under normal legislative conditions by the Iranian parliament, and some are the outcome of sociopolitical changes and special conditions (such as the Iran’s 1979 Revolution). Legislative principles of Iran’s natural resources sector include Islamic principles, basic laws, technical principles, criteria of management, policy-making, custom, tradition, culture, and international rules [14]. In order to prevent the destruction and seizure of natural resources, the first codified policy implemented in Iran was the Iran’s Forests Nationalization Law (FNL). This law was approved by the Cabinet of Iran in 1963, and by the Iranian Parliament in 1967, and was implemented from the same year (1967) [15]. Previously, individuals could acquire these resources (i.e., forests and rangelands) by seizing and converting them, and obtaining title deeds. This law brought a substantial modification to the management and legal system of Iran’s forests [16], by transferring the forest property rights to the government. Following FNL, the dependence of dwellers on forest resources, particularly in the Zagros Mountains, was not considered, and forests were managed by both traditional and governmental practices. This made the locals consider themselves the owners of the forests; yet, they faced property rights problems with the Iran’s Forestry Service. In order to apply the forest management in this region, after the FNL, conservation projects were developed since 1997. Due to their shortcomings, conservation plans were not implemented and/or were implemented only in part [17].

Iran’s Zagros forests located in western part of the country, cover around 5 million hectares [18] and play an important social, economic, and environmental role [19]; in recent decades, they experienced significant changes in cover and structure [20]. Understory farming and inappropriate land-use change are serious threats to this valuable ecosystem [21]. Unfortunately, population increase, underdevelopment consistent with the nature of the region, insufficient revenue of economic units, and employment needs in the region increased the deforestation [22]. Expanding croplands, firewood supply, and rural use have always been some of the main causes of deforestation in the Zagros oak forests [23]. Human resources in these areas are unable to provide sufficient conditions of living due to a lack of systematic attention to the agriculture sector and rural communities close to forested areas, as well as to a reduction in their contribution to economic and industrial development initiatives. This has increased cropland area and illegal seizure of forestlands; as a result, forestlands have changed to other land uses/covers (e.g., urban areas and agricultural lands).

To evaluate the performance of Governments’ conservation and management policies such as FNL and to assist the design of territorial management, regular monitoring is required to study the process and its consequences [24]. Unfortunately, there is no strategy, research, or law in Iran to assess the efficiency of FNL in expanding forests land cover. Nonetheless, remote sensing plays an important role in studying and identifying land cover changes by geospatial analysis and by providing time-related land cover maps obtained from aerial photography and satellite imageries [16]. The availability of reliable land use/cover maps is a prerequisite of monitoring forest cover change. Large-scale and accurate land use/cover mapping often requires a high degree of processing to obtain a large number of spectral-temporal characteristics that properly reflect the spectral variance of different land use/cover classes [25]. New advances, including the Google Earth Engine (GEE) computing platform, time series feature extraction methods, and machine learning algorithms, may enable a more reliable tool for large-scale forest cover change monitoring (e.g., degradation, deforestation, and recovery) [26]. A detailed understanding of forest cover dynamics may be achieved by using geospatial techniques, historical aerial photography, and long-term archives of satellite images (such as Landsat). Nita, et al. [27], for example, have used Structure from Motion (SfM) technology and historical Corona spy satellite images to estimate the effects of World War II on Romanian forest harvesting in the 1960s. SfM is a technique which may be used for land monitoring, particularly when 3D data are required [28].

The Iranian Forest Service is in charge of maintaining, rehabilitating, and improving the Zagros forests. According to the act (i.e., FNL), since 1967 this organization is also responsible for capitalizing the benefits provided by forests, meadows, and woodlands [29]. Due to the considerable changes made by the FNL of 1967 to the pre-existing property rights, there may be now different incentives for long-term sustainable use, management, and conservation. However, no studies have examined FNL-related changes in Iran’s forest cover. In this study, forest cover changes resulting from the FNL law/policy were analyzed according to management procedures for Zagros forests. For this purpose, 1955 aerial photos, 1968 Corona spy satellite photography, and satellite images (Landsat imagery) over a 68-year period (1955–2022) were used for change detection related to this law. Specifically, the study attempted to answer two research questions:How effective has the nationalization of Iran’s forests been in conserving the forested area of Zagros sparse coppice oak forest?What was the LCLUC trend in Zagros oak forests after the implementation of the FNL in Iran?

## 2. Materials and Methods

### 2.1. Study Area

Sardasht County, which is located in West Azarbaijan Province, northwest of Iran, was chosen as the study region, with a total area of around 138,000 ha (Figure 1). The county holds 90% of the province’s forests and includes four major cities (Sardasht, Mirabad, Nalas, and Rabat) and 352 rural areas [17]. The 2016’s population census revealed that urban and rural areas had a population of 68,162 and 50,687, respectively, whose main activities are agriculture and animal husbandry. The main tree species in Sardasht County are brant’s oak (*Quercus brantii*), gall oak (*Q. libani*), and aleppo oak (*Q. infectoria*) which grow in coppiced form. Other species include hawthorns (*Crataegus aronia*), common grape vine (*Vitis vinifera* L.), wild pear (*Pyrus glabra*), fig (*Ficus carica*), and wild pistachio (*Pistacia atlantica*). Sardasht’s forest stands are characterized by high (i.e., dense forests with crown cover >50%, in the northern parts of the region accounting for about 9% of the forest) to low densities (i.e., open forests with crown cover <40%).

### 2.2. Time Periods Taken into Study

The 68-year time period was divided so as to be able to provide evidence of fundamental changes brought by different policy systems. Eight time milestones for LCLUC detection (1955, 1968, 1987, 1994, 2000, 2007, 2015, and 2022) were selected based on access to appropriate aerial photos and satellite images. Accordingly, seven-time periods were chosen for analysis. The 1955–1968 time period was selected to capture the forest state before the implementation of the FNL, and the other six periods, namely 1986–1987, 1987–1994, 1994–2000, 2000–2007, 2007–2015, and 2015–2022 (covering 55 years after the implementation of the FNL), were chosen to characterize the changes after the FNL implementation.

Population density and its changes in Sardasht were collected from the Statistical Center of Iran in order to connect them to the analyzed time periods. Then, using Equation (1), the *AAG* rate (average annual growth) was computed for each period.
(1)$$AAG=\sqrt[{n}]{\frac{P_{n}}{P_{o}}}-1$$
where *n* = number of years, *P_n_* = population at the end of the period, and *P_o_* = population at the beginning of the period.

### 2.3. Data

In order to study the LCLUC and create land use/cover maps of Sardasht, the following sources were used: 1955 aerial photos (scale = 1:55,000), 1968 Corona spy satellite imagery (declassified satellite photography), and Landsat multi-temporal satellite images. The aerial photos were provided by National Geographical Organization of the Armed Forces (NGOAF) of Iran. The declassified photographic data acquired in 1968 by one Corona mission (DS1103-1041) [27] were acquired from the United States Geological Survey (USGS). This mission was implemented during the Cold War (aftermath of World War II) [30] and captured many high-resolution images which are well suited for land use/cover mapping [31].

### 2.4. Land Use Mapping

#### 2.4.1. Geo-Rectifying of Aerial Photos and Corona Satellite Photography

To create land use/cover maps of the Sardasht for 1955 and 1968, 72 pieces of aerial photos and 10 scanned, panchromatic, stereographic Corona images, combined into five pairs were analyzed. Aerial photos completely covered the study area at an average scale of 1:55,000 and they had an average coverage of 60% forward and 30% sideways. Approximately 16 × 241 km of ground is covered by each Corona film strip [32]. The aerial photos and Corona images (for an example see Figure 2) provided a full coverage of the Sardasht County at a resolution of 1.83–2.74 m.

As a first step, the geometry of the Corona images was reconstructed. The geometry of the film strips is not the same as that of aerial photographs since Corona images are normally received as a strip in four segments (a set of images): both forward and backward from the exact same place (Figure 3, top right). These frames have 75% common coverage. To decrease processing time, it was suggested to stitch each pair of photos together as two images and process them separately [33]. To provide a mosaic of images, image segments were stitched together in the Image Composite Editor 2015 (ICE) (2.0.3.0 version) software [34]. In the next step, the SfM algorithm was used in Agisoft Metashape software [35] to geo-rectify the aerial photos and stereo-pairs. First, the method of recognizing and matching image properties was used to align them. SFM matches each pixel in the photos independently of the geometric adjustments and based on the data stored in the neighboring pixel cells [36]. This algorithm detects the points that are stable in two images under different view and light conditions according to the position of the camera [37] and locates the tie points (TPs) between the photos [27]. Given that data on film parameters such as flight, camera, and image were not available for aerial photos taken in 1955 and reported by the NGOAF of Iran, in this study TPs were used to estimate the inherent parameters of the camera. Then, based on the relevant points and the camera’s inherent properties, a mosaic of orthorectified aerial photos and Corona images of Sardasht was developed.

For georeferencing the orthorectified mosaic of aerial photos, ArcGIS 10.8 software, topographic maps of 1955, vector layers of permanent rivers in the region, boundary of Sardasht and ground control points (GCPs) were used. Then, using a spline function, a mosaic of photos was georeferenced with a root mean square error (RMSE) value of less than 0.3 pixels (Figure 3). The RMSE used to estimate the positional accuracy of geospatial data, and the acquired value were appropriate based on the scale of the aerial photographs (1:55,000) [38]. Then, by visual interpretation and according to the color, tone, texture, shape, and location of features, land use/cover classes (dense forest, open forest, built-up areas, croplands, rangelands, water bodies, and barren lands; hereafter: DF, OF, BA, CL, RL, WB, and BL, respectively) were digitized in both aerial photos and Corona images. Polygons of each land use/cover class were merged and converted into raster layers with a 30 m cell size standing for the land use maps of the region in 1955 and 1968.

#### 2.4.2. Satellite Imagery

Sample extraction

Visual inspection of very high resolution satellite imagery in Google Earth (GE) was used to generate a reference dataset, which provided a sufficient number of training and validation data (samples). To obtain multi-temporal reference datasets, the sample extraction process started with the earliest time milestone (2022). For the previous ones (1987, 1994, 2000, 2007, and 2015), a forward revision process was conducted. To do this, the 2022 reference dataset was overlaid with previous ones which were in the form of GE images or/and Landsat color composites. In the next step, all the samples throughout the study area were checked and updated. The subsets for training and validation were randomly selected from the reference datasets: 70% and 30%, respectively. Table 1 lists the characteristics of the reference datasets for 2022.

Landsat time series and spectral-temporal metrics

To generate multi-temporal land use maps for all time milestones following the implementation of the FNL, Landsat satellite time series were used (Table 2). In this regard, Google Earth Engine (GEE) cloud computing platform was used to provide image collections, preprocess, extract features, classify, and run accuracy assessments of land use/cover maps. Landsat surface reflectance products from the period of 1 March to 30 October (for Landsat 9, all images were from between 1 March and 30 June) that had a cloud cover of less than 20% were used for all time points. To effectively identify the land use/cover classes, several spectral temporal metrics (STMs) including percentile metrics (5th, 25th, 50th, 75th, and 95th), standard deviation, mean, minimum and maximum of all spectral bands (B2:B7) were calculated. Along with spectral bands, several vegetation indices, namely the Soil Adjusted Vegetation Index (SAVI), Normalized Difference Vegetation Index (NDVI), Green Normalized Difference Vegetation Index (GNDVI), and Difference Vegetation Index (DVI), were calculated. In total, 90 STMs were used for land use/cover classification (Table 2). 

Landsat time series and spectral-temporal metrics

Training samples, STMs, and random forest (RF) algorithm were used for land use classification. RF is a tree-based machine learning algorithm [39] that has been widely used to classify remote sensing datasets [25]. In previous studies, RF outperformed traditional parametric (such as maximum likelihood) and novel nonparametric ML algorithms [40,41]. In GEE, several parameters can be selected and tuned to improve the learning process including but not limited to the number of trees and variables (www.developers.google.com, accessed on 2 February 2022). In this study only the number of trees for the earliest time milestone (2022) was used, and their optimal value (ntree = 100) was also used to classify land use/cover for other time milestones. Default values were used for the rest of the parameters. For the post-processing step, an iterative majority-filter tool was applied to the classified output images in ArcGIS software to simplify the last land use/cover map [42].

Using the validation samples and confusion matrices, the accuracy of thematic land use maps was assessed [43]. Accuracy metrics such as overall accuracy (*OA*), kappa coefficient, producer’s and user’s accuracies (*PA* and *UA*), *F*-score, commission and omission errors (CE and OE) were calculated based on confusion matrices [44]. The *F*-score (Equation (2)) [45] is a per-class metric computed as the harmonic mean of the user and producer’s accuracies. The *OA* and *Kappa* coefficient (Equations (3) and (4)) [46] represent the probability that a pixel is properly classified and, correspondingly, the actual agreement between reference data and the classifier used vs the chance of a random classifier.
(2)$$F-score=2\times \frac{PA\times UA}{PA+UA}$$
(3)$$OA=\frac{\sum_{i=1}^{r}X_{ii}}{N}$$
(4)$$Kappa=\frac{N\sum_{i=1}^{r}X_{ii}-\sum_{i=1}^{r}\left(X_{i+}\times X_{+i}\right)}{N^{2}-\sum_{i=1}^{r}\left(X_{i+}\times X_{+i}\right)}$$
where *PA* and *UA* are producer’s and user’s accuracies, *r* is number of rows in the confusion matrix, *X_ii_* is the number of observations in row *i* and column *i*, *X_i+_* is the total observations in row *i*, *X_+i_* is the total observations in column *i*, and *N* is the total number of observations included in the matrix.

Following adequate values for the accuracy of thematic land use maps, Equation (5) [47] was used to get the annual rate of change (*ARC*) in land use classifications for each time period:(5)$$ARC={\left[A_{2}/A_{1}\right]}^{\frac{1}{n}}-1$$
where *A*_1_ and *A*_2_ are land uses in time period one and two, and n is number of years.

### 2.5. Change Detection Analysis

The SAGA—an open-source geographic information system (GIS)—was used to run the change detection analysis. SAGA GIS contains an inbuilt tool—the confusion matrix (grid-based)—to depict changes in two land use maps [48]. This tool is useful for identification of changes in land use [49]. Then, losses and gains in each land use, net changes, persistence, and transitions from a land use to another in different land use/cover categories were calculated. The flow chart of LCLUC analysis used in this study is shown in Figure 4.

## 3. Results

### 3.1. Land Use Mapping and Accuracy Evaluation

Classification accuracy of the Landsat images was high for the created land use maps. The findings indicate that all six land use maps (1987, 1994, 2000, 2007, 2015, and 2022) were, on the whole, very accurate. The overall accuracy of the maps created for 1987, 1994, 2000, 2007, 2015, and 2022 was 89.1, 88.84, 88.75, 88.91, 89.08, and 89.02%, respectively, with Kappa coefficients of 85.06, 85.65, 85.53, 85.70, and 85.70%. Meanwhile, the PA, UA, Ce, and Oe metrics and mean F-score values for all land use/cover classes validated RF’s efficacy in land use/cover mapping (Table 3).

### 3.2. Change Detection Due to Implementation of FNL

Figure 5 and Figure 6, and Table 4 show the spatial pattern, area (ha), and change rates (%) of land use/cover for Sardasht County in the reference years (1955, 1968, 1987, 1994, 2000, 2007, 2015, and 2022). During the whole study time period (1955–2022), the area of forested lands (dense and open forests) decreased by 11,215.4 ha. The reduction of dense and open forestlands in this time period was 1511.4 and 9704.1 ha, respectively. Contrarily, there has been an increase in the area of croplands, built-up areas, and water bodies by 10,364.77, 828.5, and 1418.8 hectares, respectively. Rangelands and barren lands also experienced a decrease of 1246.9 and 109.1 ha, respectively. The results have shown that the area of Sardasht forests (dense forestlands + open forestlands) before the implementation of FNL (1955) was estimated to be 76,103.1 ha. Following the implementation of FNL, the area of forest lands had a decreasing trend, as the forest cover of 2022 accounted for 64,887.6 h. Results also indicate changes in forest lands before the implementation of the FNL (1955–1968) −1.67%. However, in the periods after the implementation of the law, the share of changes in forestlands has increased significantly. The highest rate of change (−7.86%) was recorded in the period of 1968–1987. At the expense of forestlands, the area of other land uses showed a different trend. The highest share of change in the area of croplands and built-up areas occurred in the period 1968–1987 and 1987–1994, accounting for +6.59% and +55.13%, respectively. Changes in rangelands, barren lands, and water bodies have had opposite trends. Based on the obtained results, positive (increase in area) and also negative (decrease in area) changes were observed during the study time period (1955–2022) for these three land use/cover categories.

Table 5 and Figure 7 show the results of the losses (−), gains (+), and net changes of the land use classes. As shown, most of the losses (−517 and −3006 ha) in dense and open forestlands were related to the post-FNL period (i.e., after the implementation of the law, 1968–1987). As a result, the pre-NFL period has had the lowest net changes with −497 and −119 ha. The net changes during the implementation of the law (1986–2020) increased considerably so that in contrast to the only 165 ha gains, a loss of 10,639 ha of forests (dense and open forest) was observed. Furthermore, an increase in croplands had a sustained upward trend. In the period before the implementation of the FNL (1955–1968), the net change of croplands was +578 ha, while after the implementation of the FNL, in the period 1968–1987, it increased to +2504 ha. Compared to the other two land uses, built-up areas also experienced similar net changes. Net changes of built-up areas in the period of 1955–1968 was +37 ha, and after the implementation of the FNL, since 1968, the area of these land uses has increased to +794 ha. Before implementing the FNL, no changes were observed in the rangelands. In the period of 1968–1987, the area of rangelands increased by +740 ha. But after this, the area of rangelands showed a decreasing trend. During the observed time periods, the area of water bodies had a constant trend, but due to the construction of a dam on the main river of Sardasht (Zab), their area increased by +1417 ha. The net change in barren lands after the implementation of the FNL accounted for −112 ha.

Substantial change trajectories from forestlands (dense and open forests) to other land use/cover types of the Sardasht sparse coppice oak forest are presented in Table 6 and Figure 8. During the studied periods, conversion from dense and open forests to croplands has been the most important factor in loosing forestlands. A comparison of the pattern of net changes from dense and open forests to other land uses/covers demonstrates that before the FNL (1955–1968), −119 and −498 ha of these classes’ changes were conversion to croplands. During the same period, only +1 ha of barren lands was converted to open forests. Results showed that after implementation of the FNL, −1220 and −8132 ha of dense and open forests, respectively, were converted to croplands. An increase in open forests occurred by +24 and +134 ha in the periods of 1968–1987 and 2007–2015, respectively, by conversion from dense forests.

Annual rates of changes during the studied periods (Table 7) also revealed an increased deforestation activity in Sardasht after the implementation of the FNL. Thus, the annual deforestation rate of dense forests in Sardasht increased from −0.13% in the period 1955–1968 to −0.29% and −0.92% during the periods 1968–1987 and 1987–1994, respectively. Before the FNL, net loss rate for open forestlands was −0.06%, but after implementation of the FNL this rate increased to −0.38% during 2015–2022. Croplands’ increase in area had an opposite trend to the forestlands. The annual growth rates of croplands increased from +0.13% in the period 1955–1968 to +0.62% during 1987–1994. The highest growth rate of built-up areas was in the second period after the implementation of FNL (1987–1994) by +12.13%. The rate of built-up areas development in recent years (i.e., 2015–2022) was +1.03%. The annual growth rates of rangelands also increased from +0.01% in the period 1955–1968 to +0.14% during the 1968–1987 period. Rangelands have decreased after this time period. The largest decrease (−0.64%) in the area of rangelands occurred in the period of 1994–2000. The most increase in water bodies’ area (+10.34%) occurred during 2015–2022. The largest decrease (−0.68%) in barren lands has also occurred in the same period.

### 3.3. Population Changes

Population and census data analysis has revealed that Sardasht County’s population has increased from 2667 in 1955 to 127,160 in 2022 (Figure 9). The highest average annual growth (AAG) rate of the population was estimated in the first (1955–1956) time period, accounting for 0.37.

## 4. Discussion

The nationalization of Iran’s forests in 1967 follows the prevailing philosophy of many Asian countries toward maintaining woodlands under direct governmental control, not only to protect forested lands but also to improve this resource [50]. Findings of this study make it very evident that Iran’s nationalized forest policy in the northern Zagros forests has not prevented the forest degradation and deforestation. Important parts of the policy implementation included the revision of rights to access and use forests and the preservation of genuine illegal interactions between Sardasht locals and the forest. A significant barrier to the nationalized forest management system was the clear contradiction between the legal requirements of the policies for forest protection and the actual local practices. Iran’s Forest Service had limitations in managing the Zagros forests, similar to the situation in many other nations [6,51]. Based on experiences in Latin America [52], Asia, and Africa [53,54], it is possible that forests could become free access regardless of the legal property regime if there are no institutions or procedures in place to enforce the law. In the Sardasht forests, where the Forest Service rarely monitors compliance, it appears that there is such a condition of open access. Centralized forest management by the government may not be the best strategy, especially in light of Iran’s property laws for forests and lack of local people’s participation in the decision-making process and use of indigenous knowledge in forest management plans.

The findings of this study demonstrate that, despite the FNL being implemented in Iran starting from 1967, the Northern Zagros forests were reduced more dramatically after the implementation of FNL. In this regard, dense and open forest cover changes before the FNL (1955–1968) were −0.13% and −0.06%, respectively, while after FNL these changes and the rate of decrease in forest land increased by conversion to croplands (9325 ha), built-up areas (116 ha), and rangelands (6 ha) (Table 6); if the deforestation continues, a large part of the current forest lands may be lost in the coming years. Soosani, et al. [55] used Quick Bird satellite images for a 50-year period, and found that the Central Zagros forestlands decreased by 37%. According to their study, the highest reduction in forest lands in this region was of 45% and occurred during 1955–1969, having as a main reason the requirements for wood fuel due to the cold seasons of the area between 1955 and 1969; in addition, the forest land conversion into croplands followed the nationalization of forests from 1967. Azizi Ghalati, et al. [56] noted a decrease in the amount of forest lands by 3,082 ha in the Southern Zagros over a 25-year period (1987–2012) due to the lack of adequate forest protection, extraction of wood for farm use, agriculture, livestock, and excessive grazing.

It is inferred that the context of local societies’ reliance on forest resources during these years and after the implementation of the Iran’s FNL have not been carefully reviewed. Following the implementation of FNL, local communities that previously considered themselves forest owners had difficulties in securing their resources [55]; the lack of transparency in property rights, as well as the attitude of the people towards forests as a government property has deprived people of the motivation to protect local forests, which may be an effect of implementing the FNL. Accordingly, their free access to these resources coupled with poor control and supervision from the forest organizations’ side has led to a continuous decrease in the size of these resources by conversion to croplands and built-up areas. Zandebasiri et al. [57] believe that the main contributing elements to the ineffective protection of the Zagros oak forests are the absence of specialized employment, reliance on local populations’ traditional knowledge, motivation among rural people, and unsuitable decisions among local residents. Even though this study demonstrated that Sardasht’s forests are shrinking, population growth and lack of suitable jobs, undoubtedly keep the livelihood needs high. But as the population grows and there is a greater need for employment, housing, and food, forests may find it difficult to regenerate or recover.

For the area taken into study, management plans by Iran’s Forestry Service were implemented since early 1960s, among which there are the Zagros forests conservation plans such as the coal mining organization plan [58]. Then, other plans such as exclosure (i.e., land management techniques on formerly degraded lands through protected human and livestock intervention) and regeneration, afforestation with rural households’ participation, exploitation of by-products, and preservation policies were implemented. Due to the lack of public participation, these conservation plans were not only ineffective, but they also contributed significantly to forest loss by disregarding local ownership and the lack of desire for collaborative effort in forest protection [50]. Perhaps the protection of national interests prevails over the interests of locals [59]. However, our results suggest that a proper management of Zagros forests depends on paying attention to social issues and the interaction between people and nature.

The traditional use of Zagros forests by the local people has led to a strong relationship between humans and nature in the Zagros Mountains [60]. The local people pursue a traditional management of the forests and forest products to provide fodder, firewood, and livelihood [61]. Currently, in the majority of Iran’s rural districts, villagers are in close contact with natural resources, in general, and with forests in particular, and part of their life depends on forests and natural resources [62]. Therefore, in order to properly manage the forests, using only the technical solutions and government facilities may not be the answer to the current problems faced across the country. It is important to identify employment development capacities with the collaboration of key organizations in order to improve the socio-economic structures of local populations in western Iran, so as to carry out conservation initiatives successfully, and lessen reliance on forest resources [63]. Additionally, encouraging locals to participate more could enhance forest management.

Regarding the role of the forest dwellers in the forest management process, Warner [64] suggested that organizing local people for forest harvesting operations along with forest conservation can be appropriate mechanisms in forest management. In addition, Samari and Chizari [65] have described the social forest management (people’s participation) as one of the successful approaches to forest protection. Other research [60], by documenting, gathering, and altering locals’ forest management practices in a part of Zagros region, concluded that the indigenous forest management practices (self-developed processes of forest resources management by communities) can be a short-term practice aligned with the needs of local communities, including those of fuel wood and cattle feeding [61]. Another study in this region concluded that in order to prevent the destruction of natural resources of Zagros forest, use of indigenous knowledge and customs for forests management was the best method of sustainable forest management [66]. Accordingly, one can say that with the implementation of participatory plans, natural resources can be regenerated, modified, and developed more effectively [16]. Conservation programs may be more successful when the decision-making and management for resource conservation involve local communities in the process [67,68].

## 5. Conclusions

The primary goal of this research was to acquire a better understanding of the effects of Iran’s most important forest conservation law (the FNL) on the dynamics of the Zagros forests. The following conclusions may be drawn from the study’s approach and outcomes. Historical aerial photos, Corona imagery datasets, Landsat time series, GEE-based cloud computing, spectral-temporal characteristics, and machine learning methods such as RF made large-scale forest cover monitoring possible. The results validated the study’s primary hypothesis, indicating that significant changes in the pace and trend of forest cover occurred as a result of the FNL’s implementation. Similar to some countries such as India, Bhutan, and Indonesia, based on the results of this study, Iran’s Forest Nationalization Law has not been successful in the management, protection, development, and use of forests. As a consequence, the results confirmed FNL’s ineffectiveness, demonstrating that while there was deforestation before it, the annual change in forest cover and the intensity of forest loss increased following FNL’s implementation. Hence, the law may usefully be tailored to the needs of the people, the level of knowledge and technology and policies governing society, and the reason for its formulation should be known. We may suggest a review of the policies according to the issues of the day, and to pay attention to the promotion and expansion of specialized training in order to make better use of resources. Forest management may also be improved by allowing locals to participate more. In order to guarantee appropriate planning and sustainability via decentralization and the equitable distribution of resources and opportunities, a land use planning and development system with its own council may also be useful. Ultimately, forest conservation is a global imperative and global solutions must be found and applied locally. This case study suggests various challenges and potential solutions to forest conservation for Iran and for all other nations in need of enhanced forest conservation.

## Figures and Tables

**Figure 1 sensors-23-00871-f001:**
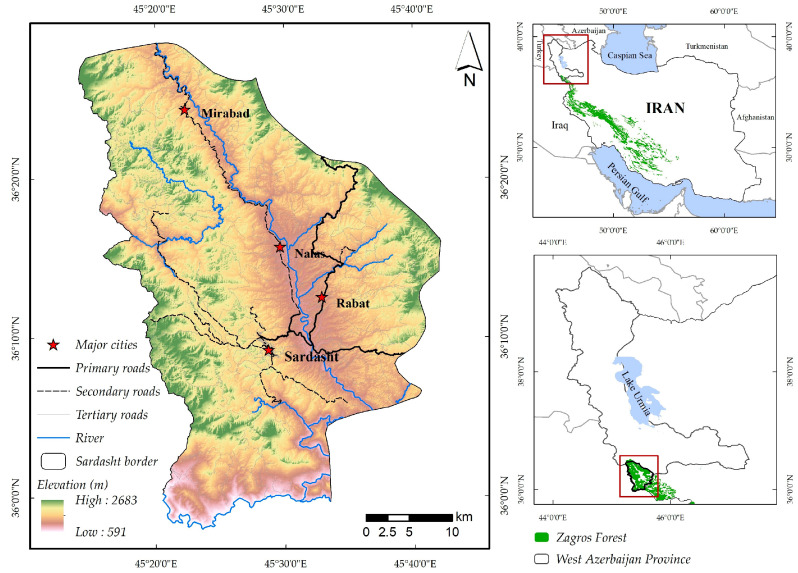
Location of the study area (Sardasht County).

**Figure 2 sensors-23-00871-f002:**
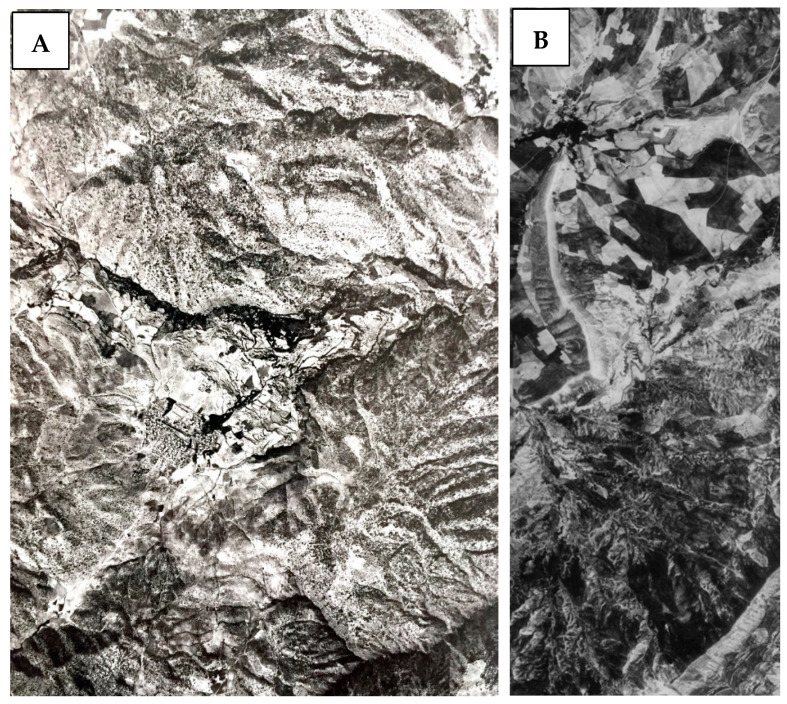
Example of a 1955 aerial photo (**A**) and a Corona film strip (**B**) in DS1103-1041 Corona Image from 4 May 1968.

**Figure 3 sensors-23-00871-f003:**
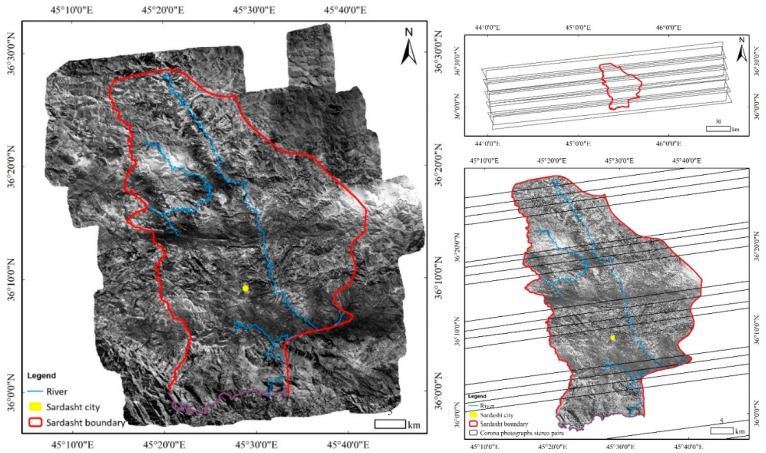
Location of Sardasht region in Corona stereo pairs (**top right**), georeferenced 1955 aerial photo (**left**) and Corona spy satellite photographs mosaic (**bottom right**).

**Figure 4 sensors-23-00871-f004:**
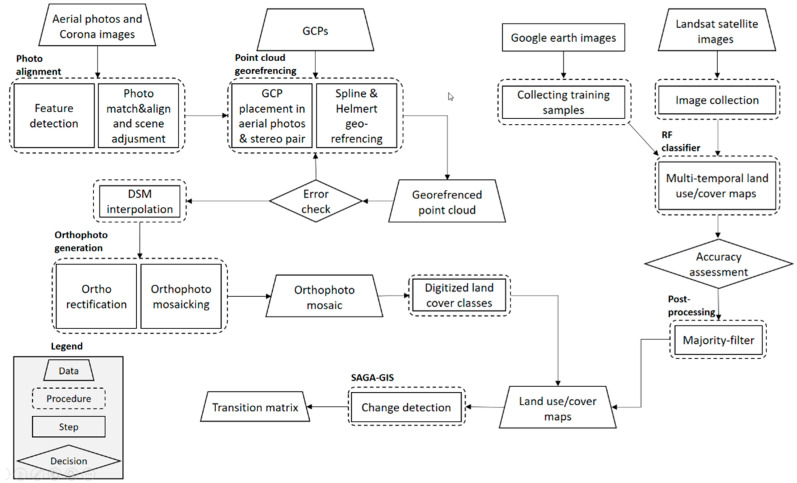
Flowchart of the data and procedures used for LCLUC detection.

**Figure 5 sensors-23-00871-f005:**
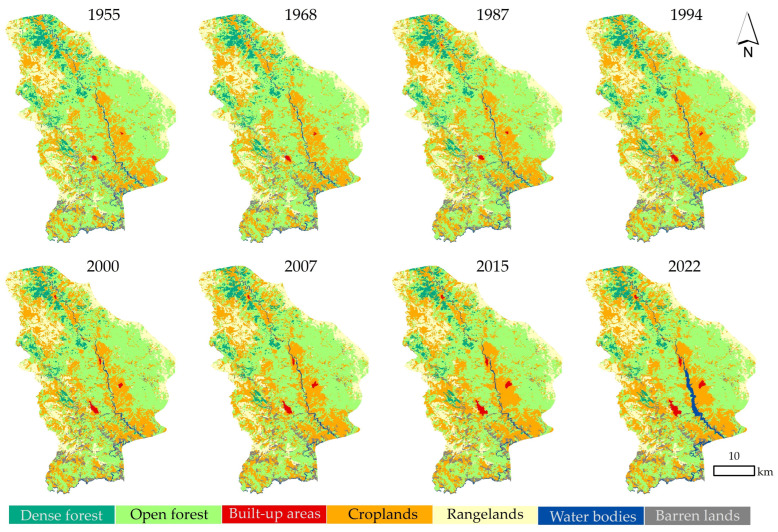
Sardasht County’s land use maps within the studied time periods.

**Figure 6 sensors-23-00871-f006:**
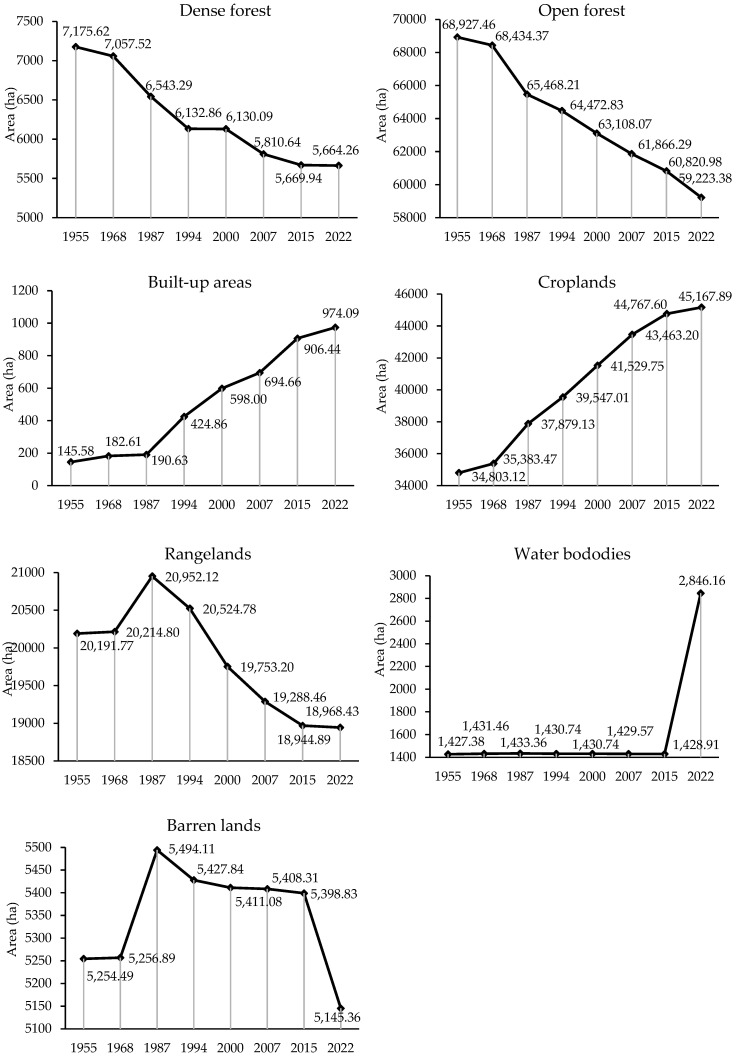
Changes in the area of land use types within the study’s time period.

**Figure 7 sensors-23-00871-f007:**
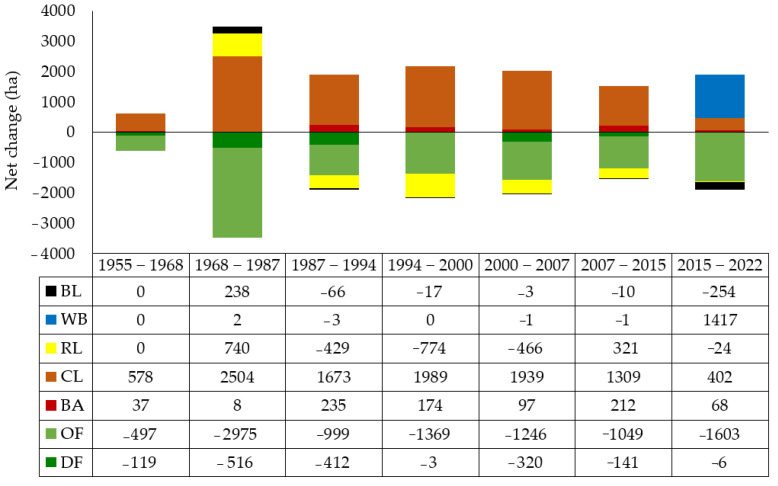
Net changes (ha) for each land use class during the studied time periods. Legend: DF—dense forest, OF—open forest, BA—built-up areas, CL—croplands, RL—rangelands, WB—waterbodies, BL—barren lands.

**Figure 8 sensors-23-00871-f008:**
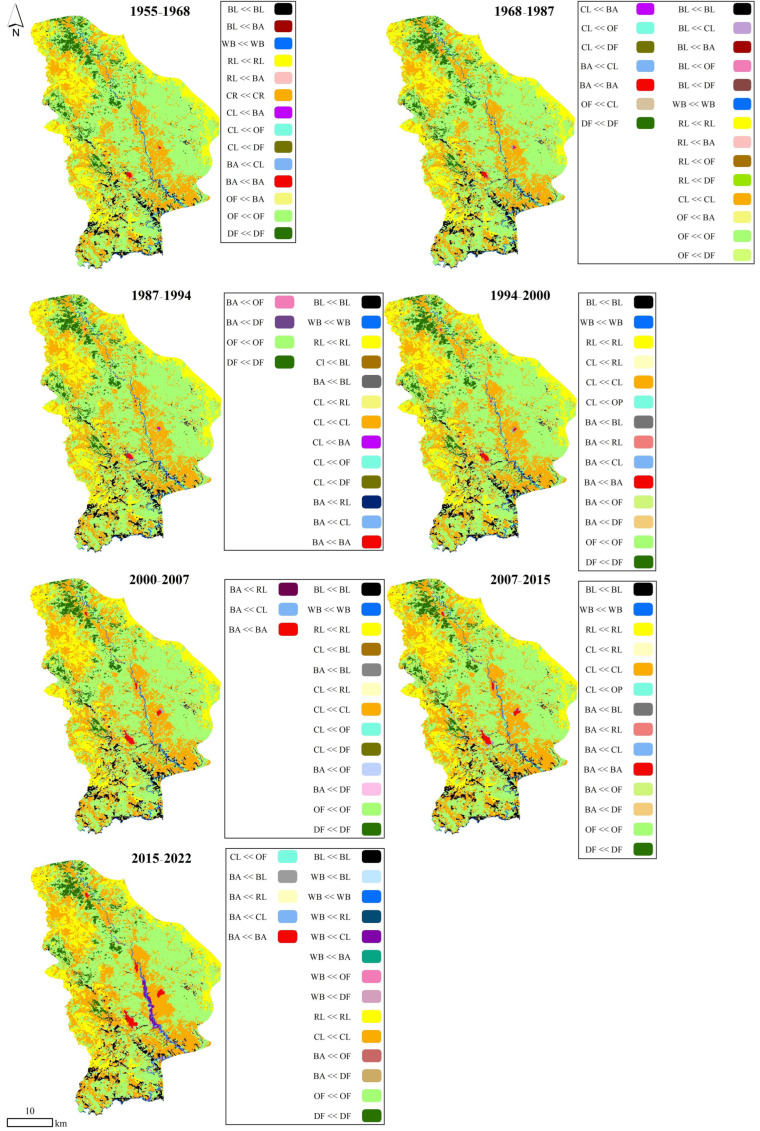
Change of land use/cover classes to other types. Legend: DF—dense forest, OF—open forest, BA—built-up areas, CL—croplands, RL—rangelands, WB—waterbodies, BL—barren lands). “<<” symbol denotes the conversion of one land use to another.

**Figure 9 sensors-23-00871-f009:**
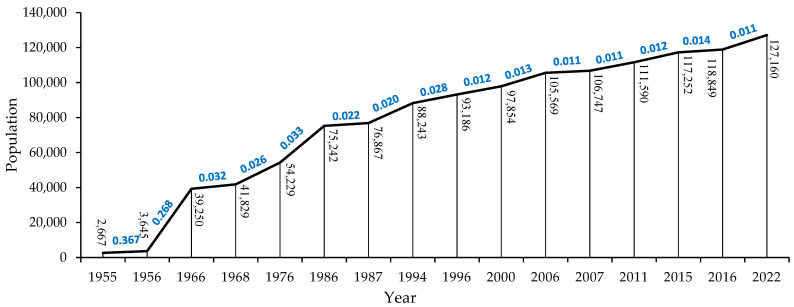
Population statistics for the census years and studied time periods in Sardasht County. Numbers in bold and blue color indicate average annual growth (AAG) rate of population between different years.

**Table 1 sensors-23-00871-t001:** Details of the reference dataset for 2022: the number of samples and pixels used for training and validation subsets.

Land Use/Cover Class	Training Samples		Validation Samples	
	No. of Samples	No. of Pixels (10 m)	No. of Samples	No. of Pixels (10 m)
DF	135	4839	57	2073
OF	316	11,384	135	4888
BA	25	1036	12	444
CL	340	1353	147	595
RL	238	880	102	480
WB	245	875	105	375
BL	28	1034	13	442

Land use classes: DF—dense forest, OF—open forest, BA—built-up areas, CL—croplands, RL—rangelands, WB—waterbodies, and BL—barren lands.

**Table 2 sensors-23-00871-t002:** Details of used Landsat imagery and STMs for multi-temporal land use/cover classification.

Time Point	Data Type	Sensor	No. of Images	STMs	No. of STMs
2022	Landsat 9	Operational Land Imager-2 (OLI-2)	19	Percentile metrics (5th, 25th, 50th, 75th, and 95th) + standard deviation + mean + minimum + maximum of spectral bands and vegetation indices	90
2015	Landsat 8	Operational Land Imager (OLI)	29		
2007	Landsat 5	Thematic Mapper (TM)	15		
2000	Landsat 7	Enhanced Thematic Mapper Plus (ETM+)	10		
1994	Landsat 5	Thematic Mapper (TM)	13		
1987	Landsat 5	Thematic Mapper (TM)	18		

**Table 3 sensors-23-00871-t003:** Image classification accuracy evaluation for the studied years.

Land Use/Cover Class	1987					1994					2000				
	PA	UA	F	CE	OE	PA	UA	F	CE	OE	PA	UA	F	CE	OE
DF	89.21	88.18	88.69	10.78	11.81	87.21	85.94	86.57	12.79	12.77	94.49	91.79	93.12	5.51	8.21
OF	90.83	83.43	86.97	9.17	16.56	91.05	83.67	87.20	8.95	16.32	89.71	86.49	88.07	10.28	13.50
BA	69.09	74.09	71.50	49.10	25.90	87.38	77.91	82.37	14.21	24.55	92.34	85.06	88.55	7.66	14.94
CL	90.34	93.85	92.06	9.65	6.14	91.41	94.08	92.73	8.59	5.92	91.29	91.74	91.51	8.71	8.25
RL	89.37	93.84	91.55	10.62	6.16	87.57	91.13	89.31	12.42	8.87	87.02	89.14	88.07	12.98	10.86
WB	97.26	92.87	95.01	2.74	7.13	91.04	96.56	93.72	27.94	3.43	81.26	78.25	79.73	18.74	21.97
BL	86.95	88.51	87.72	13.04	11.49	68.11	80.8	73.91	34.89	19.15	64.38	87.11	74.04	35.61	12.89
**Land Use/Cover Class**	**2007**					**2015**					**2022**				
	**PA**	**UA**	**F**	**CE**	**OE**	**PA**	**UA**	**F**	**CE**	**OE**	**PA**	**UA**	**F**	**CE**	**OE**
DF	88.19	92.81	90.44	11.81	7.19	95.24	90.91	93.02	3.77	8.77	95.24	92.68	93.94	4.76	7.32
OF	92.02	82.45	86.97	7.98	17.55	86.69	87.53	87.11	13.34	10.74	86.9	88.27	87.58	13.10	11.72
BA	79.95	96.42	87.42	20.99	11.14	91.46	90.86	91.16	9.20	11.14	87.74	98.52	92.82	12.25	1.47
CL	91.38	92.66	92.02	8.62	7.34	89.26	89.24	89.25	10.98	10.91	87.5	89.07	88.28	12.50	10.93
RL	86.18	91.62	88.82	13.81	8.37	91.97	91.2	91.58	8.85	10.73	91.31	89.39	90.34	8.68	10.61
WB	88.47	88.25	88.36	11.53	11.75	89.1	83.48	86.20	9.95	16.71	95.91	86.41	90.91	4.09	13.58
BL	71.01	85.85	77.73	28.99	14.14	64.77	87.42	74.41	23.08	7.38	76.46	78.69	77.56	23.54	21.30

Land use/cover classes: DF—dense forest, OF—open forest, BA—built-up areas, CL—croplands, RL—rangelands, WB—waterbodies, and BL—barren lands. PA and UA—Producer’s and User’s accuracies (%), F—F-score (%), CE and OE—Commission and Omission errors (%).

**Table 4 sensors-23-00871-t004:** Land use/cover classes change rates between the studied time periods.

Land Use/Cover Class	Time Periods						
	1955–1968	1986–1987	1987–1994	1994–2000	2000–2007	2007–2015	2015–2022
DF	−1.67%	−7.86%	−6.69%	−0.05%	−5.50%	−2.48%	−0.10%
OF	−0.72%	−4.53%	−1.54%	−2.16%	−2.01%	−1.72%	−2.70%
BA	20.28%	4.21%	55.13%	28.95%	13.92%	23.36%	6.95%
CL	1.64%	6.59%	4.22%	4.77%	4.45%	2.91%	0.89%
RL	0.11%	3.52%	−2.08%	−3.91%	−2.41%	−1.69%	−0.12%
WB	0.29%	0.13%	−0.18%	0.00%	−0.08%	−0.05%	49.80%
BL	0.05%	4.32%	−1.22%	−0.31%	−0.05%	−0.18%	−4.93%

Land use/cover classes: DF: dense forest, OF: open forest, BA: built-up areas, CL: croplands, RL: rangelands, WB: waterbodies, and BL: barren lands.

**Table 5 sensors-23-00871-t005:** Losses (−) and gains (+) (ha) in land use classes across the studied time periods.

Land Use/Cover Class	Time Periods													
	1955–1968		1986–1987		1987–1994		1994–2000		2000–2007		2007–2015		2015–2022	
	Loss	Gain	Loss	Gain	Loss	Gain	Loss	Gain	Loss	Gain	Loss	Gain	Loss	Gain
DF	−119	0	−517	1	−412	0	−3	0	−320	0	−141	0	−6	0
OF	−498	1	−3006	30	−999	0	−1369	0	−1246	0	−1182	134	−1603	0
BA	−7	44	−47	55	−46	281	0	174	0	97	−1	213	−19	86
CL	−44	622	−124	2627	−124	1797	−138	2127	−89	2029	−170	1479	−927	1328
RL	0	0	0	740	−429	0	−774	0	−466	0	−321	0	−24	0
WB	0	0	0	2	−3	0	0	0	−1	0	−1	0	0	1417
BL	0	0	0	238	−66	0	−17	0	−3	0	−10	0	−254	0

Land use/cover classes: DF—dense forest, OF—open forest, BA—built-up areas, CL—croplands, RL—rangelands, WB—waterbodies, and BL—barren lands.

**Table 6 sensors-23-00871-t006:** Major change trajectories (ha) from forestlands (dense and open forests) to other types of land use classes in studied time periods.

	Time Periods		DF	OF	CL	RL	WB	BA
Pre-NFL	1955–1968	DF	-	-	−119	-	-	-
		OF	-	-	−498	-	-	+1
	Total	DF	-	-	−119	-	-	-
		OF	-	-	−498	-	-	+1
Post-NFL	1968–1987	DF	-	−24	−484	−6	−1	-
		OF	+24	-	−2103	−731	−1	+1
	1987–1994	DF	-	-	−410	-	-	−2
		OF	-	-	−954	-	-	−44
	1994–2000	DF	-	-	-	-	-	−3
		OF	-	-	−1358	-	-	−11
	2000–2007	DF	-	-	−320	-	-	−1
		OF	-	-	−1241	-	-	−4
	2007–2015	DF	-	−134	−6			−1
		OF	+134	-	−1148	-	-	−34
	2015–2022	DF	−6	-	-	-	-	-
		OF	-	-	-1328	-	−257	−17
	Total	DF	−6	−158	−1220	−6	−1	−7
		OF	+268	-	−8132	−731	−258	−109

Land use/cover classes: DF: dense forest, OF: open forest, BA: built-up areas, CL: croplands, RL: rangelands, WB: waterbodies, and BL: barren lands.

**Table 7 sensors-23-00871-t007:** Annual rate of change (ARC) and its percent in different land use/cover types during the studied time periods.

Time Periods		Land Use/Cover Class						
		DF	OF	BA	CL	RL	WB	BL
1955–1968	ARC	−0.0013	−0.0013	−0.0013	−0.0013	−0.0013	−0.0013	−0.0013
	%ARC	−0.13	−0.13	−0.13	−0.13	−0.13	−0.13	−0.13
1968–1987	ARC	−0.0029	−0.0029	−0.0029	−0.0029	−0.0029	−0.0029	−0.0029
	%ARC	−0.29	−0.29	−0.29	−0.29	−0.29	−0.29	−0.29
1987–1994	ARC	−0.0092	−0.0092	−0.0092	−0.0092	−0.0092	−0.0092	−0.0092
	%ARC	−0.92	−0.92	−0.92	−0.92	−0.92	−0.92	−0.92
1994–2000	ARC	−0.0001	−0.0001	−0.0001	−0.0001	−0.0001	−0.0001	−0.0001
	%ARC	−0.01	−0.01	−0.01	−0.01	−0.01	−0.01	−0.01
2000–2007	ARC	−0.0076	−0.0076	−0.0076	−0.0076	−0.0076	−0.0076	−0.0076
	%ARC	−0.76	−0.76	−0.76	−0.76	−0.76	−0.76	−0.76
2007–2015	ARC	−0.0031	−0.0031	−0.0031	−0.0031	−0.0031	−0.0031	−0.0031
	%ARC	−0.31	−0.31	−0.31	−0.31	−0.31	−0.31	−0.31
2015–2022	ARC	−0.0001	−0.0001	−0.0001	−0.0001	−0.0001	−0.0001	−0.0001
	%ARC	−0.01	−0.01	−0.01	−0.01	−0.01	−0.01	−0.01

Land use/cover classes: DF—dense forest, OF—open forest, BA—built-up areas, CL—croplands, RL—rangelands, WB—waterbodies, and BL—barren lands.

## Data Availability

Not applicable.

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
