# Peer review of "Impact of Iran’s Forest Nationalization Law on Forest Cover Changes over Six Decades: A Case Study of a Zagros Sparse Coppice Oak Forest"

_sensors, 2023, doi:10.3390/s23020871_

Round 1

Reviewer 1 Report

In this manuscript, the authors are interested in the impact of Iran’s forest nationalization (FNL) law on forest cover changes over a period of 60 years. They collect a large number of aerial images and process them with the aim of answering two questions:

1) How effective has the nationalization of Iran's forests been in conserving the forested area of Zagros sparse coppice oak forest?

2) What was the LCUCC trend in Zagros oak forests after the implementation of the FNL in Iran?

 Similar to other related studies, the authors of the manuscript reach the conclusion that Iran's Forest Nationalization Law has not been successful in the management, protection, development and use of forests.

Although this is a rather lengthy manuscript, spanning 23 pages, nevertheless, it is relatively well-written. The introduction primes the reader to what the manuscript is about and provides the needed context with relevant, yet not so recent references. The materials and methods section describes the area in Iran that is under study, the time periods considered, the land use mapping as well as the satellite imagery employed. Relevant tables, figures and mathematical formula ease in reading comprehension. The results section is well laid out and the computed and tabulated results are mostly clear (Figure 5 seems to be cropped from the right side though). The discussion section analyzes the results and does so adequately. The manuscript is finalized with a conclusions section that is based on the earlier results and their analysis and discussion.

 Overall, this is a well-written manuscript. I only have minor comments:

1. Figure 5 is cropped.

2. Figure 8 is of low resolution.

Author Response

Many thanks to the respected reviewer for her/his kind notes. We tried to consider the point ‎that the respected reviewer mentioned which showed with green highlight in the text.‎ In addition, figures 5 and 8 were updated in response to your request.

Reviewer 2 Report

It is better to use use LULC instead of LCLC, because it is more common in the references.

Introduction section must be modified since there is no section related to the background of algorithms and datasets which are used in similar studies.

NW Iran? Do you mean north-west?

Please add coordinate system to Figure 1.

Which criteria is used for selection of 1955, 1968, 1987, 1994, 2000, 2007, 2015, and 2022 in this study?

Which method is used for georeferencing?

Ask a native to revise your paper. some sentences need to be modified. For example: In GEE, several tuning 240 parameters can be defined to improve the learning process including but not only how 241 many trees and variables there are?

In addition to Kappa and OA, you need more accuracy criteria such as F-score to prove efficieny of your methodolgy.

In figure 5, some texts are missed because of bad alignment.

Is area continuous in the figure 6?

Conclusion must be modified.

Author Response

Many thanks to the respected reviewer for her/his kind notes. We tried to consider the point that the respected reviewer mentioned which was addressed with yellow highlight in the text. Responses to your comments are also provided here.

Respected reviewer comments:

It is better to use LULC instead of LCLC, because it is more common in the references.

Response:

Dear reviewer, we have checked the text several times, but we did not find a word like “LCLC”. We used the “official” Land-Cover and Land-Use Change (LCLUC) ‎in the manuscript which is defined by NASA. The following references are examples of this term:

Justice, C., Gutman, G., & Vadrevu, K. P. (2015). NASA land cover and land use change (LCLUC): An interdisciplinary research program. Journal of Environmental Management, 148, 4-9. Cited by 89

Turner, B. L., & Geoghegan, J. (2004). Land-Cover and Land-Use Change (LCLUC) in the Southern Yucatán Peninsular Region (SYPR). In People and the Environment (pp. 31-60). Springer, Boston, MA. Cited by 12

Green, K., Kempka, D., & Lackey, L. (1994). Using remote sensing to detect and monitor land-cover and land-use change. Photogrammetric engineering and remote sensing, 60(3), 331-337. Cited by 556

Rogan, J., & Chen, D. (2004). Remote sensing technology for mapping and monitoring land-cover and land-use change. Progress in planning, 61(4), 301-325. Cited by 851

López, E., Bocco, G., Mendoza, M., & Duhau, E. (2001). Predicting land-cover and land-use change in the urban fringe: A case in Morelia city, Mexico. Landscape and urban planning, 55(4), 271-285. Cited by 851

Sleeter, B. M., Loveland, T., Domke, G., Herold, N., Wickham, J., & Wood, N. (2018). Land cover and land-use change. In: Reidmiller, DR; Avery, CW; Easterling, DR; Kunkel, KE; Lewis, KLM; Maycock, TK; Stewart, BC, eds. 2018. Impacts, Risks, and Adaptation in the United States: Fourth National Climate Assessment, Volume II. Washington, DC: US Global Change Research Program. pp. 202–231., 2, 202-231.

Introduction section must be modified since there is no section related to the background of algorithms and datasets which are used in similar studies.

Response:

Noted and introduction was amended.

NW Iran? Do you mean north-west?

Response:

Noted and the text modified‎. We're referring to Iran's northwest.

Please add coordinate system to Figure 1.

Response:

Noted and the coordinate systems added to figure.

Which criteria is used for selection of 1955, 1968, 1987, 1994, 2000, 2007, 2015, and 2022 in this study?

Response:

The time periods were chosen based on the date of ratification of Iran's forest nationalization law. We wanted to look at how land use/cover changed before and after implementation of this law. Access to suitable aerial photos and cloud-free satellite imagery to change detection was also a key factor in selecting these time points.

Which method is used for georeferencing?

Response:

We utilized spline transformation in ArcGIS 10.5 software to determine the correct map coordinate location for each cell in the raster (i.e., aerial photos).

Ask a native to revise your paper. some sentences need to be modified. For example: In GEE, several tuning 240 parameters can be defined to improve the learning process including but not only how 241 many trees and variables there are?

Response:

We appreciate your feedback. The text has been corrected, and two natives have edited it. The modifications they made are noted in yellow.

In addition to Kappa and OA, you need more accuracy criteria such as F-score to prove efficiency of your methodology.

Response:

Your stated new accuracy evaluation technique (F-score) was developed and put into practice.

In figure 5, some texts are missed because of bad alignment.

Response:

Noted and the figure has been updated.

Is area continuous in the figure 6?

Response:

The position of the figure in the text was changed.

Conclusion must be modified.

Response:

Noted and conclusion was amended.

Round 2

Reviewer 2 Report

In my perspective, this paper can be accepted in its present form.